# Transferring Textual Knowledge for Visual Recognition

## Abstract

Transferring knowledge from task-agnostic pre-trained deep models for down-stream tasks is an important topic in computer vision research. Along with the growth of computational capacity, we now have open-source Vision-Language pre-trained models in large scales of the model architecture and amount of data. In this study, we focus on transferring knowledge for vision classification tasks. Conventional methods randomly initialize the linear classifier head for vision classification, but they leave the usage of the text encoder for downstream visual recognition tasks undiscovered. In this paper, we revise the role of the linear classifier and replace the classifier with the embedded language representations of the object categories. These language representations are initialized from the text encoder of the vision-language pre-trained model to further utilize its well-pretrained language model parameters. The empirical study shows that our method improves both the performance and the training speed of video classification, with a negligible change in the model. In particular, our paradigm achieves the state-of-the-art accuracy of 87.3% on Kinetics-400.

## 1 Introduction

Pre-training a task-agnostic model using large-scale general datasets and then transferring its learning feature representations to downstream tasks is a paradigm in many computer vision applications [1, 2]. While in the last decade, the convolutional-based models that are optimized on the ImageNet [3] (more precisely, ILSVRC-2012) dataset with a supervised style dominated this field. Owing to the dramatically increasing computational capacity, now we can train models that have several magnitude more model parameters and FLOPs on significantly larger datasets in either supervised [4, 2, 5], weakly-supervised [1, 6] or self-supervised [7, 8] style. Recently, contrastive learning-based vision-language pre-training [1] manifest their superior capabilities in improving down-streaming tasks performance such as classification [1], captioning [9], image generation [10, 11], to name a few. These models are powerful for two reasons: i) the employed large-scale weakly-related datasets provide rich semantics and diverse representations of concepts; ii) the representation vectors of images and texts are roughly aligned in the semantic embedding space. However, the most common approach to using these models is fine-tuning the visual encoder on specific tasks. Although the rich semantics and diverse representations of concepts benefit the downstream tasks, the usage of the textual encoder is still left undiscovered.

In this study, we aim to improve the transferability of such vision-language pre-training models for downstream classification tasks, with the help of their textual encoders. Our motivation comes from the semantic similarity among the ground-truth labels. To demonstrate this, we employ the kinetics video recognition dataset [12] for the analysis. We extract the embedded textual vectors of class labels using the textual encoder released by CLIP [1]. We then calculate the correlation between the

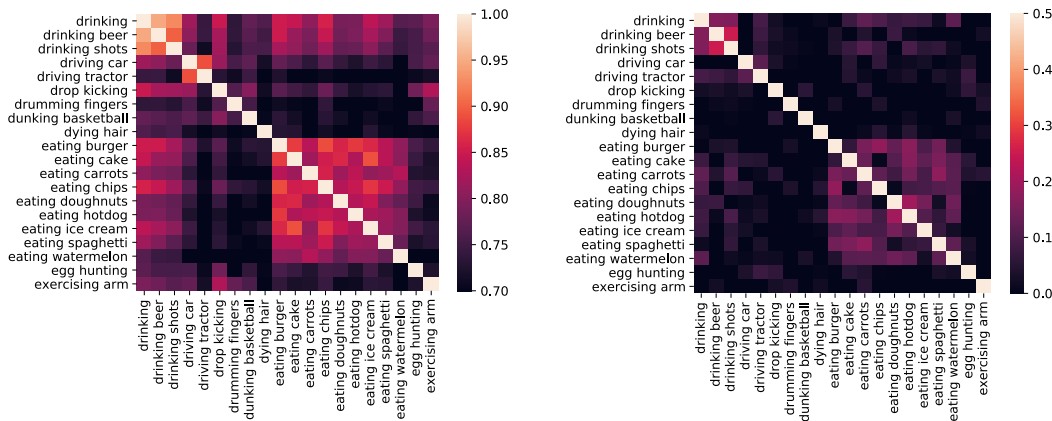

Figure 1: Inter-class correlation maps of "embeddings of class labels" for 20 categories on Kinetics-400. **Left:** The extracted textual vectors of class labels, **Right:** The "embeddings" from learned classifier. The color thresholds are adjusted for better understandability. Please zoom in for best view.

embedded textual vectors. The plot is shown on the left of Figure 1. Not surprisingly, the extracted textual vectors of class labels exhibit certain inter-class correlations, since part of them include the same verbs in their labels, such as *playing <something>*. Meanwhile, the labels with different verbs show a negligible inter-class correlation, such as *drinking* and *driving*. Next, we examine the final projection head of a vanilla visual recognition framework. We conduct the visual-only fine-tuning progress with the visual encoder that is also released by CLIP [1]. The detailed configurations are provided in Section 4.2. The projection head is a matrix of $d \times c$ to compute the pre-softmax values (or logits) from the $d$-dimensional feature vectors for the $c$ classes. Non-rigorously, we can consider the $d$-dimensional row vectors as the embeddings of the class labels, allowing us to explore the inter-class correlation between these learned "embeddings", as shown on the right side of Figure 1. Interestingly, these learned "embeddings" also reveal certain correlations after the training progress, despite being initialized randomly and optimized without knowing any textual information [1].

Therefore, we suppose that the semantic information contained in the samples (images and videos) does correlate with inter-classes. Following this motivation, we replace the projection matrix with several variants: i) A projection matrix whose row vectors are randomly sampled (trivial correlation); ii) A projection matrix whose row vectors are orthogonal to each other (non-correlated). Then we replace the projection matrix with fixed embedded textual vectors that provide the "proper" correlation. In the empirical studies, we find that the textual knowledge significantly improves the transferability of pre-trained models, regarding both the classification accuracy and the convergence speed. Our main contributions are summarized as follows:

- We build a new recognition paradigm to improve the transferability using knowledge from the textual encoder of the well-pretrained vision-language model.

- We conduct extensive experiments on popular video and image datasets (*i.e.*, Kinetics-400 [12], UCF-101 [13], HMDB-51 [14] and ImageNet [3]) to demonstrate the transferability of our solution in many types of transfer learning, *i.e.*, image/video recognition, zero-shot recognition, few-shot recognition. Our approach democratizes the training on large-scale video/image datasets and achieves state-of-the-art performance on video recognition tasks, *e.g.*, 87.3% top-1 accuracy on Kinetics-400.

## 2   Methodology

**Denotations.** In the rest of the paper, we use bold letters to denote `Vector`, and capital italic letters to denote `Tensor` or `Matrix`. For instance, we employ $\mathbf{z} \in \mathbb{R}^d$ to denote the feature vector extracted from a pre-trained model of dimension $d$, we employ $W \in \mathbb{R}^{d \times c}$ to denote the projection matrix for the $c-$class linear classifier. Without ambiguity, we also use capital italic letters to denote the

---
[1]That is, optimized with cross-entropy loss with one-hot labels

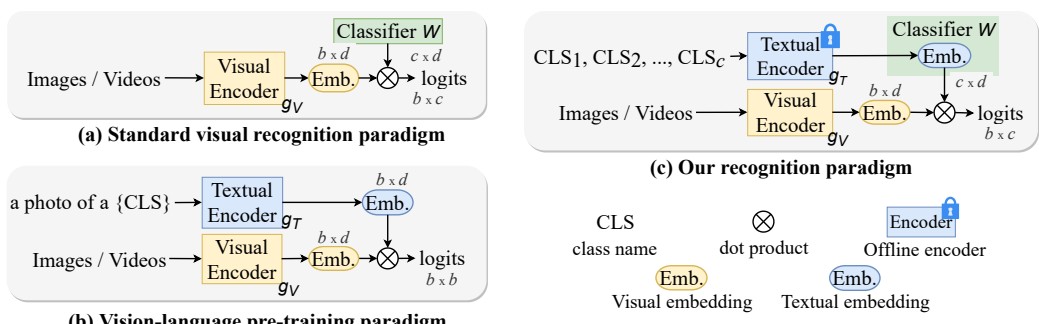

Figure 2: Illustration of (a) standard visual recognition paradigm, (b) vision-language pre-training paradigm, and (c) our proposed recognition paradigm.

modality in subscripts, especially we employ $V$ and $T$ to denote the *Visual* modality and *Textual* modality, respectively. We further employ lowercase italic letters to denote functions or neural networks. For instance, we employ $g_V(\cdot, \Theta_V)$ and $g_T(\cdot, \Theta_T)$ to denote the visual encoder and textual encoder, respectively. Additionally, we employ calligraphic letters, *e.g.*, $\mathcal{D}$, to denote sets of elements.

## 2.1 Revisiting of the standard paradigm and the vision-language pre-training

**Standard visual feature transferring paradigm.** We start with the most ordinary scenario, where a visual feature encoder model $g_V$ is optimized using a large-scale dataset $\mathcal{D}$ that contains visual samples with or without ground-truth labels. On our labeled downstream dataset $\tilde{\mathcal{D}} = \{(\boldsymbol{x}_1, \boldsymbol{y}_1), (\boldsymbol{x}_2, \boldsymbol{y}_2), \ldots\}$, our empirical learning target can be written as

$$g_V^*, W^* = \underset{\Theta_V, W}{\operatorname{argmin}} \, \mathbb{E}_{\boldsymbol{x}, \boldsymbol{y} \sim \tilde{\mathcal{D}}} \big[ H(\boldsymbol{y} | \sigma(W \cdot g_V(\boldsymbol{x}))) \big], \tag{1}$$

where $H(\hat{p}|p)$ stands for the `CrossEntropy` between the predicted distribution $p$ and the ground-truth distribution $\hat{p}$, $\sigma$ denotes the `softmax` operation, $W \in \mathbb{R}^{c \times d}$ denotes the linear projection matrix for classification. The formulation in Eq. 1 is a standard visual feature transferring paradigm, where the visual encoder $g_V$ and the projection matrix (classifier) $W$ are learned simultaneously.

**Vision-language pre-training in CLIP.** As shown in Figure 2(b), we then review the contrastive pre-training paradigm of the vision-language models in [1]. Given a weakly related image-text pair dataset $\mathcal{D} = \{(\boldsymbol{x}_{V,1}, \boldsymbol{x}_{T,1}), (\boldsymbol{x}_{V,2}, \boldsymbol{x}_{T,2})...\}$. With slight abuse of the notations, we employ the $\boldsymbol{x}_V, \boldsymbol{x}_T$ to denote a mini-batch of size $b$, then we minimize the following target,

$$g_V^*, g_T^* = \underset{\Theta_V, \Theta_T}{\operatorname{argmin}} \, \mathbb{E}_{\boldsymbol{x}_V, \boldsymbol{x}_T \sim \tilde{\mathcal{D}}} \big[ H(\mathcal{Q} | \sigma(g_V(\boldsymbol{x}_V)^{\mathrm{T}} \cdot g_T(\boldsymbol{x}_T))) \big], \tag{2}$$

where $\mathcal{Q}$ is the set that contains $b$ one-hot labels of size $c$, with their $1, 2, \ldots, b$ -th element being $1$ ($b < c$, denoting the positive image-text pairs. Here we clarify that, the definition in Eq. 2 is not the rigorous form of the Noise-Contrastive Estimation (NCE) loss proposed in [15, 16]. Instead, we employ the cross entropy version implementation in [1, 17]. This implementation depicts a connection between the standard feature transferring paradigm and ours. In which, the $g_T(\boldsymbol{x}_T)$ can be considered as the projection matrix that map the visual feature $g_V(\boldsymbol{x}_V)$ to the given label set $\mathcal{Q}$.

## 2.2 Our proposed paradigm

As discussed in Section 1, we replace the learnable randomly initialized linear projection matrix $W$ with pre-defined matrix $\tilde{W}$. Similarly, the training target can be written as

$$g_V^* = \underset{\Theta_V}{\operatorname{argmin}} \, \mathbb{E}_{\boldsymbol{x}, \boldsymbol{y} \sim \tilde{\mathcal{D}}} \big[ H(\boldsymbol{y} | \sigma(\tilde{W} \cdot g_V(\boldsymbol{x}))) \big]. \tag{3}$$

Note that $\tilde{W}$ is not in the optimization targets, since we freeze it from updating during the fine-tuning on the downstream tasks. We do this for two reasons: Firstly, it could preserve the textual knowledge from being disturbed by the randomness brought by the mini-batch. For instance, when some classes are missing, their embedded feature vector might be broken by the other classes; Secondly, we want

to provide a fair comparison between different initializations of $\tilde{W}$ (The unfrozen results are given in the supplementary materials). Now we consider how to initialize $\tilde{W}$. To examine how the correlation between the semantic information contained in the samples helps, we investigate the following four types of initialization, where the forth is our proposed initialization.

**Randomized matrix** For the most simple randomized matrix case, we set each row of the $\tilde{W}$ with a random Gaussian vector of zero mean and standard deviation, that is

$$\tilde{W} \sim \mathcal{N}(\mathbf{0}, I_d), \tag{4}$$

where $I_d$ denotes the identity matrix of dimension $d \times d$. Arithmetically, a trivial "correlation" would appear between the row of the $\tilde{W}$, since the sampling size is significantly small to be biased. Evidently, the trivial "correlation" cannot indicate the real correspondence between the classes due to its stochasticity. Therefore we expect the model to have inferior performance since it needs to avoid these incorrect correlations when learning the visual feature representation.

**Randomized Orthogonal matrix** We follow the approach of the randomized matrix. We then remove the correlation by ensuring the row vectors are orthogonal. This is achieved by QR decomposition. Concretely, since $d > c$, we first generate a random matrix of size $d \times d$ and select the first $c$ rows as our projection matrix. Formally, we have,

$$\tilde{W}_j \sim \mathrm{QR}(U)_j, j = 1, 2, \ldots, c, \quad U_i \sim \mathcal{N}(\mathbf{0}, I_d), i = 1, 2, \ldots, d, \tag{5}$$

where $U$ is the intermediate randomized matrix, $\mathrm{QR}(U)$ is the row orthogonal matrix obtained through the QR decomposition. Similar to the randomized matrix, we also expect this initialization to have inferior performance. Given the fact that the one-hot label vectors are also orthogonal to each other, it will not be helpful to project the visual feature vectors with an orthogonal matrix, which increases the difficulty of learning meaningful visual features.

**Linear discriminant projection** We consider another way of initializing the projection matrix. We employ the multi-class Fisher's linear discriminant analysis (LDA) to learn a linear classifier, then employ the weight matrix of the classifier as our initialization of the projection matrix. The LDA is optimized using the visual embeddings from the pre-trained model of samples in the train split. Then we compute the projection matrix following previous work [18]. Intuitively, the LDA first projects the feature vectors into a lower dimension space that maximizes the inter-class covariance and then estimates the likelihood of a sample to the class distributions. We, therefore, term this as the maximal correlation initialization. As an essential classifier, this type of initialization delivers reasonable performance, but it is largely dependent on the data employed to compute the projection matrix. When the data is limited, the estimated correlation will be biased. On the other hand, in our proposed paradigm, the pre-trained textual encoder provides unbiased correlations for fine-tuning.

**Textual embedding vectors** We finally describe our proposed feature transferring paradigm. Briefly, the projection weight $\tilde{W}$ is composed of the embedded textual feature vectors of the labels. Given a set of tokenized class labels $\mathcal{L} = \{l_1, l_2, \ldots, l_c\}$, we have

$$\tilde{W}_i \sim g_T(l_i), i = 1, 2, \ldots, c, \tag{6}$$

where $\tilde{W}_i$ the $i$-th row vector in matrix $\tilde{W}$. And $\tilde{W}_i$ is initialized using the textual encoder output of the textual label of the $i$-th class. In the experimental analysis, we investigate two types of textual feature encoders: i) The encoder that is trained with a visual encoder in the contrastive style; ii) The encoder that is trained solely using only textual samples on tasks such as masked language modeling.

## 3 Related Works

**Visual Recognition.** Convolutional networks have long been the standard for backbone architectures in image recognition [19, 20, 21, 22, 23, 24] and video recognition [25, 26, 27, 28, 29, 30, 31]. Inspired by the Transformer [32] scaling successes in Natural Language Processing, Vision Transformer (ViT) [33] applies a standard Transformer directly to images, which delivers impressive performance on image recognition. Since then, ViT [33] has led a new trend in image recognition backbone architectures, shifting from CNNs to Transformers. To improve performance, follow-up studies (*e.g.*, DeiT [34], Swin [35]) have been developed. Also, many works has begun to adopt transformers in video recognition, such as TimeSFormer [36], ViViT [37], VideoSwin [38], and MViT [39].

**Vision-language Pre-training.** Recently, CLIP [1] provides good practice in learning the coordinated vision-language pre-training models using the image-text InfoNCE contrastive loss [40]. Based on CLIP, several variants [41, 42, 43, 44, 45] have been proposed by combining more types of learning tasks such as image-text matching and masked image/language modeling. These contrastively learned models have two deserved properties for downstream tasks: the abundant visual feature representations and the aligned textual feature representations. Yet another study [46] merged the downstream classification task into the pre-training progress, which demonstrates a decent improvement of accuracy over the standard cross-entropy loss. Moreover, a few recent works [47, 48] transfer the CLIP [1] pre-trained image-text matching model to the downstream video-text matching framework for video recognition with contrastive loss. Specifically, ActionClip [47] extends the CLIP [1] to train a downstream video-text matching model and then perform video recognition indirectly using the similarity between learned video and text encoders during inference. [48] focus on efficient prompting and learning the continuous prompt template as text input for video recognition. Instead of these matching-based approaches, we aim to propose a new recognition paradigm that directly transfers textual knowledge for visual recognition. Our approach can balance performance and efficiency, and experiments demonstrate that our approach can reduce computational power requirements while democratizing training on large-scale video/image datasets (see Table 6 and 12 for more information).

# 4 Experiments: Video Recognition

## 4.1 Setups

To evaluate our method for video recognition, we conduct experiments on three widely used benchmarks, *i.e.*, Kinetics-400 [12], UCF-101 [13] and HMDB-51 [14]. See Supp. for more details.

**Training & Inference.** We utilize ResNet [20] and ViT [33] as the visual encoders since they are the representative backbones of CNN and vision transformer, respectively. We employ the pre-trained visual and textual encoder released by CLIP [1] in most experiments for simplicity. Given a video, we first uniformly sampled $T$ (*e.g.*, 8, 16, 32) frames over the entire video. Then image patches with the resolution of 224×224 are randomly cropped from the sampled frames to form the input. The model is optimized using AdamW with momentum set to 0.9. We use an initial learning rate of $5e^{-6}$, a cosine learning rate schedule with a 5-epoch linear warmup and a batch size of 128 for experiments on all datasets. For fast training, we set the total training epoch to 30 unless specified otherwise.

To trade off accuracy and speed, we consider two evaluation protocols. (1) *Single View*: We use only 1 clip per video and the center 224×224 crop for efficient evaluation, (*e.g.*, as in Section 4.2). (2) *Multiple Views*: This is a widely used setting in previous works [49, 27, 50] to sample multiple clips per video (*e.g.*, 10 clips) with several spatial crops (*e.g.*, 3 crops) in order to get higher accuracy. For comparison with SOTAs, we use four clips with three 224×224 crops ("4×3 Views") in Table 7.

## 4.2 Ablations on Kinetics.

In this section, we conduct extensive ablation experiments to demonstrate our method with the instantiation. Models in this section use 8-frame input, ViT-B/16 as the visual backbone, 30 epochs for training and a single view for testing on Kinetics-400, unless specified otherwise.

**Comparison with vision-only framework.** Figure 2(a) illustrates the standard visual recognition framework. As a comparison with our method, we train the unimodality video model, which consists of the same visual encoder and a learnable classifier with random initialization. To produce video embedding, we just apply temporal average pooling (TAP) to frame embeddings. As presented in Figure 3, our method surpasses *Vision-Only* baselines across multiple label fractions on Kinetics-400. Especially when just only 10% labeled data is available for training, demonstrating that the advantage of our paradigm is more profound when the labeled data is limited. Also, when training with full data, our *Vision-Text* method leads to an additional 5% improvement with the same training recipe. Figure 4 further demonstrates our paradigm significantly improves convergence speed.

**Different assignments to the offline classifier.** We set different initializations described in section 2.2 to the offline classifier $W \in \mathbb{R}^{d \times c}$ and then train our visual encoder on Kinetics-400. Table 1 lists their comparisons. We show that feeding the offline classifier a random $d$-by-$c$ matrix with a normal distribution reduces performance significantly. Then we assign the orthogonal matrix to the classifier,

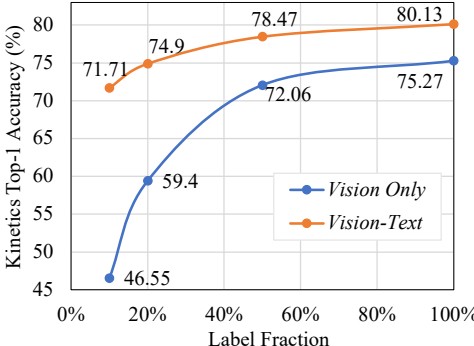 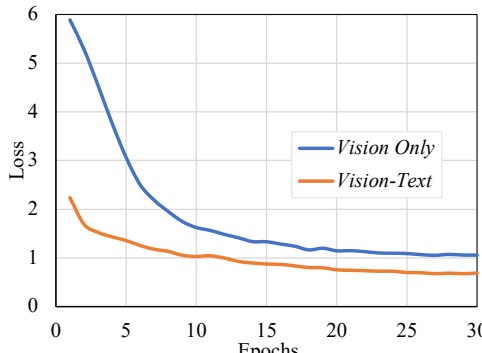

Figure 3: Vision-Text *v.s.*Vision-only framework under different label fractions on Kinetics-400.

Figure 4: The training loss of Vision-Text and Vision-only framework on Kinetics-400.

and we can see that having different classes that are orthogonal will result in inferior performance. Also, we choose DistilBERT [51] as the textual encoder to pre-extract the text embeddings of $c$ categories. The resulting performance is the same as that of the CLIP's textual encoder. Furthermore, we term the linear discriminate projection as the maximal correlation initialization, as stated in Section 2.2. To do so, we first sample 60 videos from each class in the training set and utilize the pre-trained visual encoder to extract visual embeddings from these 24,000 videos. Finally, we learn the linear classifier by performing linear discriminant analysis on these visual embeddings and their ground-truth labels. We can see that the result of the LDA projection is consistent with our statement. More visualizations of these classifiers are in supplementary materials.

Table 1: Exploration of different generation methods for the frozen classifier.

| Offline classifier from | Top 1 |
| --- | --- |
| Textual encoder of CLIP | 81.52 |
| Random normal matrix | 59.30 |
| Random orthogonal matrix | 59.44 |
| DistilBERT | 81.45 |
| Linear discriminant projection | 80.77 |

Table 2: Temporal modeling for video encoders.

| Backbone | Modeling | Top-1 | Top-5 |
| --- | --- | --- | --- |
| | TAP | 71.20 | 90.37 |
| ResNet-50 | T1D | 67.18 | 88.45 |
| | T-Trans | 74.26 | 91.67 |
| | TAP | 80.13 | 94.98 |
| VIT-B/16 | TokenT1D | 80.42 | 95.03 |
| | T-Trans | 81.52 | 95.49 |

**Temporal modeling.** Here we explore more temporal modelings for ViT [33] and ResNet [20]: (1) **TAP**: Temporal average pooling is the most straightforward temporal modeling. (2) **T1D**: The channel-wise temporal 1D convolutions, is a common strategy [50, 52, 53], to perform efficient temporal interaction in the latter stages (*i.e.*, res$_{4-5}$) of ResNet. (3) **T-Trans**: The embeddings of frames are fed to a multi-layer (*e.g.*, 6-layer) temporal transformer encoder. (4) **TokenT1D**: We use T1D to model temporal relations for [class] token features that are aggregated from local features via attention in the vision transformer. We perform the TokenT1D in multiple positions of a vision transformer. Results are shown in Table 2. On both backbones, TAP provides simple baselines and T-Trans exhibits the best top-1 accuracy. Both of them maintain the original frame-level representations and then perform temporal modeling. An interesting thing we observed is that T1D does not seem to work in this scenario. The reason lies in that T1D may have the potential to break the learned strong representations provided by CLIP. TokenT1D is another internal-backbone temporal modeling, and it does not yield a performance drop, and even slightly improves the TAP baseline. We believe this is because TokenT1D is only imposed on the global [class] token features instead of patches features, resulting in minimal modifications on pre-trained features.

**Visual encoder with different pre-training.** Besides CLIP-pretrained visual encoders, we further explore our paradigm with different pre-trained visual encoders. As shown in Table 3, equipped with ImageNet-pretrained visual encoder, our method helps to improve the vision-only counterpart by 0.9%. We can see that the CLIP-pretrained visual encoder achieves more significant performance, which is probably because CLIP provides the coarse initial alignment between frames and category names, as well as covers rich visual concepts.

**Text input forms.** Intuitively, the name of a class appears to be the most straightforward text information. We can see that only using the label text can yield good results in Table 4. Then

Table 3: Study on different pre-training.

| Visual encoder | Paradigm | Top-1 |
|---|---|---|
| CLIP-pretrained | Vision-Only | 75.27 |
| | Vision-Text | 80.13 |
| ImageNet-pretrained | Vision-Only | 74.78 |
| | Vision-Text | 75.63 |

Table 4: Study on various text input forms.

| Text input from | Top 1 |
|---|---|
| class name | 81.37 |
| "a video of a person" + class name | 81.52 |
| multiple fixed templates + class name | 80.88 |
| learnable template + class name | 81.22 |

following the prompt engineering in CLIP [1], we utilize the prompt template "a video of a person {label}." to help specify the text is about the content of the video. This only slightly increases performance over the baseline of using the label text. We further use multiple prompt templates as the text augmentation during training. Performance decreases by 0.64% on Kinetics-400. This may be because different prompt templates may introduce extra noise for the training. In addition to the hand-crafted prompt, we also adopt an automated prompt [54] to describe a prompt's context using a set of learnable vectors. The results suggest that different templates have little impact on our model.

Table 5: Different instantiations of our method on Kinetics-400. "Single View" indicates one temporal clip with one spatial crop, whereas "4×3 Views" indicates 4 temporal clips with 3 spatial crops.

| Encoder | Resolution | Frames | Single View | | 4×3 Views | |
|---|---|---|---|---|---|---|
| | | | Top-1 | Top-5 | Top-1 | Top-5 |
| ResNet-50 | 224×224 | 8 | 74.26 | 91.67 | 75.50 | 92.61 |
| | | 16 | 74.81 | 92.20 | 75.94 | 93.00 |
| VIT-B/32 | 224×224 | 8 | 77.97 | 93.80 | 79.57 | 94.70 |
| | | 16 | 79.17 | 94.24 | 80.37 | 94.95 |
| VIT-B/16 | 224×224 | 8 | 81.52 | 95.49 | 82.65 | 96.25 |
| | | 16 | 82.34 | 95.71 | 83.15 | 96.25 |
| VIT-L/14 | 224×224 | 8 | 84.82 | 96.59 | 85.83 | 97.05 |
| | | 16 | 85.85 | 96.47 | 86.36 | 96.88 |
| | | 32 | 86.39 | 96.75 | 87.09 | 97.06 |
| VIT-L/14 | 336×336 | 8 | 84.94 | 96.55 | 86.23 | 97.11 |
| | | 16 | 86.05 | 96.92 | 86.63 | 97.27 |
| | | 32 | 86.60 | 97.00 | 87.30 | 97.46 |

**More instantiations.** We assess different instantiations of our paradigm, in terms of different visual encoders, more input frames, and larger spatial resolution. See Supp. for more details on architectures. In Table 5, we present the results of our method with two typical evaluation protocols. In general, more frames, larger spatial resolution, and deeper backbones lead to higher accuracy.

Table 6: Ours *vs.* Matching paradigm with ViT-B/16 on Kinetics-400. The number of V100-days is the number of V100 GPU used for training multiplied by the training time in days. ∗ indicates the official result [47] via "Data-parallel training" on 3090 GPUs. For efficient training and fair comparison, we implement all experiments with "Distributed Data-parallel training" in the Table.

| Method | Batch gather | Textual encoder | Top-1 | Top-5 | V100-days |
|---|---|---|---|---|---|
| Matching paradigm [47] | ✓ | online | 81.15 | 95.42 | 6.7 (10*) |
| | ✓ | offline | 80.73 | 95.36 | 6.6 |
| | ✗ | online | 77.77 | 94.79 | 3.5 |
| | ✗ | offline | 76.13 | 94.57 | 3.3 |
| Our paradigm | ✗ | offline | 81.52 | 95.49 | 3.3 |

**Our recognition paradigm *vs.* Matching paradigm.** Here we make a comparison with the matching-based method mentioned in Section 3. The matching paradigm treats the recognition task as a video-text matching problem with contrastive loss, thus requiring a batch gathering to collect embeddings of all batches across all GPUs and calculate cosine similarity for a given batch across all other batches. See Supp. for details about the batch gathering. In Table 6, we try to compare

Table 7: Comparison to SOTAs on Kinetics-400. "Views" indicates # temporal clip $\times$ # spatial crop. The magnitudes are Giga ($10^9$) and Mega ($10^6$) for FLOPs and Param. "IN" denotes ImageNet.

| Method | Input | Pre-train | Top-1 | Top-5 | FLOPs×Views | Param |
|---|---|---|---|---|---|---|
| NL I3D-101 [27] | $128\times224^2$ | IN-1K | 77.7 | 93.3 | $359\times10\times3$ | 61.8 |
| MVFNet$_{En}$ [50] | $24\times224^2$ | IN-1K | 79.1 | 93.8 | $188\times10\times3$ | - |
| SlowFast NL101 [49] | $16\times224^2$ | Scratch | 79.8 | 93.9 | $234\times10\times3$ | 59.9 |
| X3D-XXL [55] | $16\times440^2$ | Scratch | 80.4 | 94.6 | $144\times10\times3$ | 20.3 |
| MViT-B, $64\times3$ [39] | $64\times224^2$ | Scratch | 81.2 | 95.1 | $455\times3\times3$ | 36.6 |
| *Methods with large-scale pre-training* | | | | | | |
| TimeSformer-L [36] | $96\times224^2$ | IN-21K | 80.7 | 94.7 | $2380\times1\times3$ | 121.4 |
| ViViT-L/16×2 [37] | $32\times320^2$ | IN-21K | 81.3 | 94.7 | $3992\times4\times3$ | 310.8 |
| Swin-L [38] | $32\times384^2$ | IN-21K | 84.9 | 96.7 | $2107\times10\times5$ | 200.0 |
| ip-CSN-152 [56] | $32\times224^2$ | IG-65M | 82.5 | 95.3 | $109\times10\times3$ | 32.8 |
| ViViT-L/16×2 [37] | $32\times320^2$ | JFT-300M | 83.5 | 95.5 | $3992\times4\times3$ | 310.8 |
| ViViT-H/16×2 [37] | $32\times224^2$ | JFT-300M | 84.8 | 95.8 | $8316\times4\times3$ | 647.5 |
| TokLearner-L/10 [57] | $32\times224^2$ | JFT-300M | 85.4 | 96.3 | $4076\times4\times3$ | 450 |
| MTV-H [58] | $32\times224^2$ | JFT-300M | 85.8 | 96.6 | $3706\times4\times3$ | - |
| CoVeR [59] | $16\times448^2$ | JFT-300M | 86.3 | - | $-\times1\times3$ | - |
| Florence [44] | $32\times384^2$ | FLD-900M | 86.5 | 97.3 | $-\times4\times3$ | 647 |
| CoVeR [59] | $16\times448^2$ | JFT-3B | 87.2 | - | $-\times1\times3$ | - |
| Ours ViT-L/14 | $32\times224^2$ | WIT-400M | 87.1 | 97.1 | $1662\times4\times3$ | 230.7 |
| Ours ViT-L/14 | $32\times336^2$ | WIT-400M | 87.3 | 97.5 | $3829\times4\times3$ | 230.7 |

with the matching paradigm [47] as fairly as we can. We can see that the matching paradigm does not work well without batch gather. This is due to contrastive learning favors a large batch size. Besides, involving batch gather will multiply the training time. Also, in this case, the pre-trained textual encoder still needs to be updated, which requires larger GPU memory. However, our paradigm employs pre-extracted text embeddings as our classifier, so the only thing we need to fine-tune is the visual encoder. Results show that our method achieves the best accuracy-cost trade-off. Specifically, our method achieves the performance of 81.52% with VIT-B/16, which takes only 10 hours to run the training using 8 GPUs (2×faster than the matching counterpart).

### 4.3 Main Results.

**Comparison to state-of-the-art.** In Table 7, on Kinetics-400, we compare to state-of-the-arts that are pre-trained on large-scale datasets such as ImageNet-21K [3], IG-65M [60], JFT-300M [2], FLD-900M [44] and JFT-3B [5]. The suffix represents the magnitude of the dataset, *e.g.*, JFT-3B consists of nearly 3 billion annotated images. We include the details of these web-scale datasets in Supp. To the best of our knowledge, up to now, none of the three largest datasets (*i.e.*, JFT-300M, FLD-900M, JFT-3B) are open-sourced and also do not provide pre-trained models. Thus, we use the CLIP [1] checkpoints, which are publicly available[2] and have been trained on 400 million web image-text pairs (namely WIT-400M). Observe that we achieve state-of-the-art results. Specifically, our model outperforms all JFT300M-pretrained methods in terms of Top-1 and Top-5 accuracy. We achieve 87.3%, which improves even further by 0.8% over Florence [44], although their model and data scale are both 2×larger. Besides, our model is even better than JFT3B-pretrained CoVeR [59], and their data scale is 7.5×larger. See Supp. for more results on UCF-101 and HMDB-51 datasets.

**Few-shot video recognition.** Video recognition using only a few samples is known as few-shot video recognition. We study a more challenging $K$-shot $C$-way situation instead of the conventional 5-shot 5-way configuration. We scale the task up to categorize **all** categories in the dataset with just $K$ samples per category for training. The upper bound of this situation is denoted by the term "All-shot". Table 8 reports the top-1 accuracy for the three datasets. In this extreme scenario of few data, we use 200 epochs to train models with ViT-B/16 for few-shot video recognition. For temporal modeling, we use TAP. We can observe that our method provides amazing transferability on diverse domain data in these extreme data-poor circumstances.

---

[2]https://github.com/openai/CLIP/blob/main/clip/clip.py

Table 8: Few-shot video recognition on three popular datasets under $K$-shot $C$-way setting.

| K-shot | K400 | UCF101 | HMDB51 |
|--------|-------|--------|--------|
| 1 | 63.16 | 88.77 | 65.17 |
| 3 | 67.50 | 92.78 | 69.99 |
| 5 | 69.89 | 93.87 | 71.03 |
| All | 80.13 | 95.24 | 73.18 |

Table 9: Zero-shot video recognition under intra-dataset and cross-dataset settings. {A}→{B} indicates we train the model on dataset A then perform zero-shot recognition on dataset B.

| | K300→K100 | K400→UCF |
|--------------|-----------|----------|
| Ours w/o train | 63.35 | 63.01 |
| Ours w/ train | 66.38 | 74.67 |

**Zero-shot video recognition.** We conduct experiments on two open-set settings: 1) Intra-dataset: The Kinetics-400 was divided into two parts: 300 categories (K300) for training and 100 categories (K100) for zero-shot recognition. 2) Cross-dataset: We train our models on K400 and then evaluate them on UCF101. To avoid catastrophic forgetting [61], here we train our models with few epochs. As shown in Table 9, unlike the traditional recognition paradigm, ours can achieve zero-shot recognition for unseen categories by replacing the offline classifiers. Appropriately tweaking the pre-trained model slightly can boost performance even further.

# 5 Experiments: Image Recognition

We also evaluate our approach to the image recognition task. Here we conduct experiments on ImageNet [3] and share the same training recipe in section 4.1 with ImageNet.

**Few-shot image recognition.** Here we also use the challenging $K$-shot $C$-way setting on ImageNet. Specifically, the models are trained using $K$ images (shots) from the training set for each image category and then measure performance on the corresponding standard 1000-class testing set. As shown in Table 10, the results reveal that our method has strong transferability under data-poor conditions, whereas the standard unimodality paradigm is ineffective in comparison to ours.

Table 10: Few-shot image recognition on ImageNet. "Zero-shot" and "All-shot" denote the lower and upper bounds of the task respectively. Top-1 accuracy is reported here.

| K-shot | 0 | 1 | 3 | 5 | All |
|--------|-------|-------|-------|-------|-------|
| Ours | 66.73 | 71.50 | 73.64 | 74.99 | 82.25 |
| Vision-Only | 0 | 4.71 | 30.44 | 41.70 | 79.70 |

Table 11: Zero-shot image recognition. We train the model on IN600 then perform evaluation on IN400.

| | IN600→IN400 |
|--------------|-------------|
| Ours w/o train | 70.28 |
| Ours w/ train | 72.62 |

**Zero-shot image recognition.** Here we split the ImageNet-1K into two parts, with 600 categories (IN600) for training, and the remaining unseen 400 categories (IN400) for evaluation. Table 11 demonstrates the zero-shot image recognition ability of our method.

**Efficient training.** For readers' reference, we provide the performance of our approach with different visual backbones on ImageNet in Tabel 12. Notably, using 8 GPUs, we can train the VIT-B/16 to achieve 82.25% in 90 minutes, while the ViT-L/14 only takes 6 hours to achieve 86.47%.

Table 12: Study on various backbones. Models are trained with 10 epochs.

| Backbone | Resolution | Top-1 | Top-5 | FLOPs | Params | A100-days |
|----------|------------|-------|-------|--------|--------|-----------|
| VIT-B/16 | 224×224 | 82.25 | 96.82 | 11.3G | 57.3M | 0.5 |
| VIT-L/14 | 224×224 | 86.47 | 98.11 | 51.9G | 202.1M | 2.0 |
| VIT-L/14 | 336×336 | 87.12 | 98.33 | 116.5G | 202.1M | 5.7 |

# 6 Conclusion

We present a new paradigm for improving the transferability of visual recognition that is based on the knowledge from the textual encoder of the well-trained vision-language model. The empirical study shows that our method improves both the performance and the convergence speed of visual classification. The proposed approach has superior performance on both general and zero-shot/few-shot recognition and achieves state-of-the-art performance on video recognition tasks, and democratizes training on large-scale video/image datasets.

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
