# Supplementary for
# Transferring Textual Knowledge
# for Visual Recognition

## 1 A Additional Results on Video Recognition

### 2 A.1 Video datasets

3 • **Kinetics-400** (K400) [1] is a large-scale video dataset, which consists of 240k training
4 videos and 20k validation videos in 400 different human action categories.

5 • **UCF-101** [2] contains 13k videos spanning over 101 human actions.

6 • **HMDB-51** [3] contains approximately 7k videos belonging to 51 action class categories.

### 7 A.2 Comparison with state-of-the-arts on UCF-101 and HMDB-51

8 We also evaluate our method on the UCF-101 and HMDB-51 datasets to demonstrate its capacity to
9 generalize to smaller datasets. We finetune our models on these two datasets using the pre-trained
10 ViT-L model on Kinetics-400 and present the mean class accuracy over three splits utilizing 8 frames
11 as inputs and 30 epochs for training. Table 1 reveals that our model has a pretty transfer capability,
12 with mean class accuracy of 98.2% on UCF-101 and 79.0% on HMDB-51, respectively.

Table 1: **Mean class accuracy** on UCF-101 and HMDB-51 achieved by different methods which are transferred from their **Kinetics** models with RGB modality (over 3 splits).

| Method | UCF-101 | HMDB-51 |
|---|---|---|
| $ECO_{En}$ [4] | 94.8% | 72.4% |
| ARTNet [5] | 94.3% | 70.9% |
| I3D [6] | 95.6% | 74.8% |
| R(2+1)D [7] | 96.8% | 74.5% |
| S3D-G [8] | 96.8% | 75.9% |
| TSM [9] | 95.9% | 73.5% |
| STM [10] | 96.2% | 72.2% |
| TEINet [11] | 96.7% | 72.1% |
| MVFNet [12] | 96.6% | 75.7% |
| TDN [13] | 97.4% | 76.4% |
| Ours | **98.2**% | **79.0**% |

### 13 A.3 More visualizations of different classifiers

14 Here we provide more visualizations of different classifiers in Figure 1.

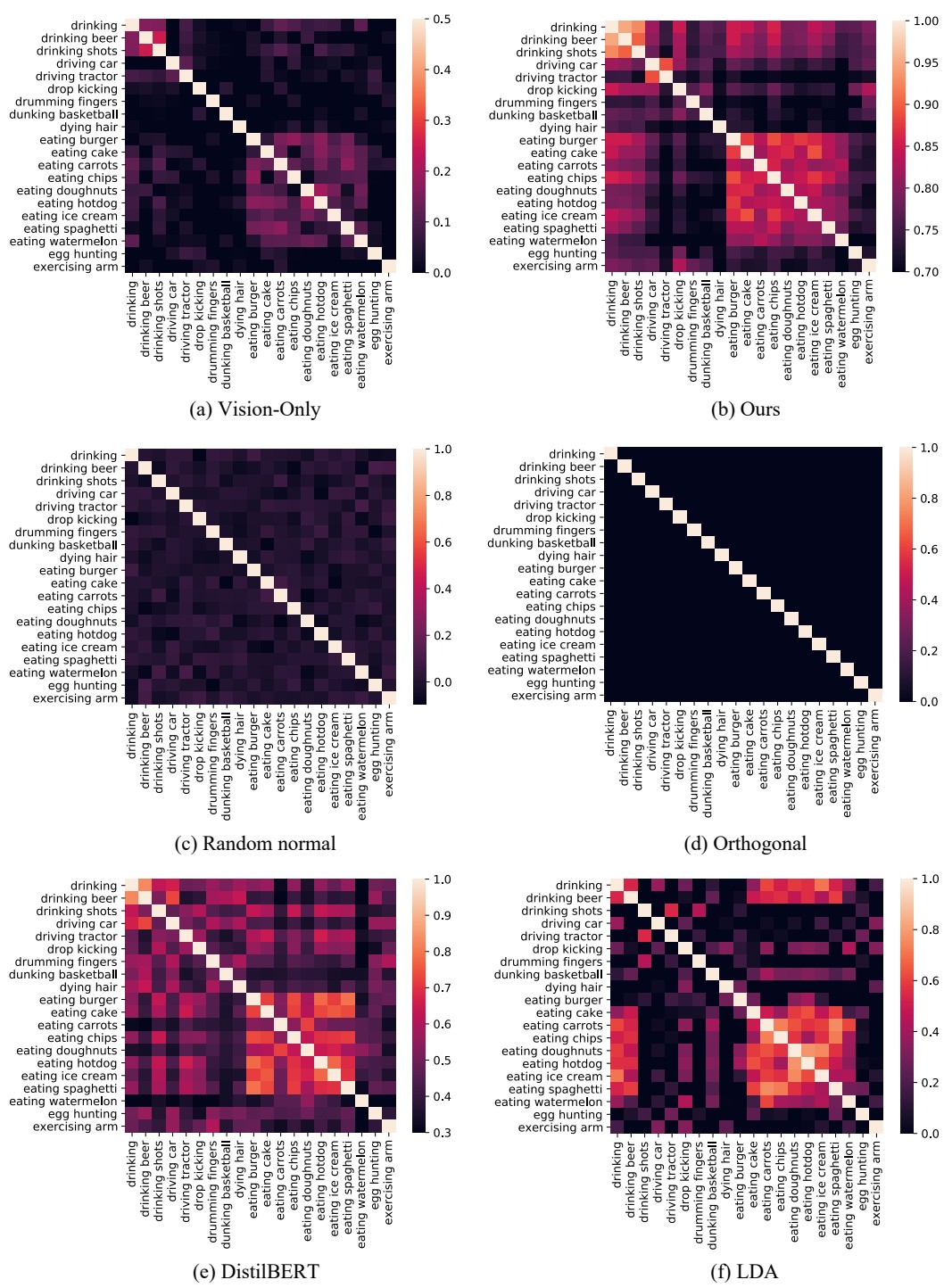

Figure 1: Inter-class correlation maps of "embeddings of class labels" for 20 categories on Kinetics-400. The color thresholds are adjusted for better understandability. Please zoom in for best view.

### A.4 Comparison with the unfrozen classifier

As we described in Section 2.2 of the submission, we freeze the classifier from updating during the fine-tuning of the downstream tasks for the reason: It could preserve the textual knowledge from being disturbed by the randomness brought by the mini-batch. By doing so, we can replace the offline classifier and do zero-shot recognition.

Here, we test the unfrozen classifier with the same textual embeddings as the frozen classifier. The unfrozen results are given in Table 2. We can see that the unfrozen setting causes the original textual knowledge to be broken, resulting in a decrease in performance.

Table 2: Frozen classifier *vs.* Unfrozen classifier.

| Setting | Top-1 | Top-5 |
|---------|-------|-------|
| frozen | 81.52 | 95.49 |
| unfrozen | 79.16 | 93.55 |

## B Additional Implementation Details

### B.1 Large-scale datasets for pre-training

Here we describe the large-scale web-scale datasets used in other video recognition methods for pre-training. The suffix of the name represents the magnitude of the dataset.

- **ImageNet-1K/21K**: The ImageNet-1K dataset was used to pre-train models for computer vision transfer learning. It was first released for the ILSVRC2012 visual recognition challenge. The ImageNet-1K dataset is a subset of the larger ImageNet dataset, which contains 14,197,122 images split into 21,841 categories. The whole dataset is known to as ImageNet-21K (sometimes referred to as ImageNet-22K) and has been **open-source** [1]. ImageNet-1K was created by selecting a subset of 1.2M images from ImageNet-21K, that belong to 1000 mutually exclusive classes.

- **IG-65M**: Facebook has proposed the IG-65M dataset, which contains approximately 65 million public, user-generated Instagram videos with hashtags. Due to label and temporal noise, the dataset is used for weakly-supervised training. This dataset is not open-source, but several pre-trained R(2+1)D [7] and CSN [14] models are **provided** [2].

- **JFT-300M**: JFT-300M is an internal Google dataset used to train image classification models. The dataset consists of 300M images that are labeled with 18,291 categories. Image labels are generated using a complex algorithm that combines raw web signals, web page connections, and user feedback. However, the dataset and the pre-trained weights are **not** open-source.

- **FLD-900M**: FLD-900M is a large image-caption dataset from Microsoft, which includes 900M Images and 900M Free form text (From one word, Phrase to sentence). By now, the dataset and the pre-trained weights are **not** open-source.

- **JFT-3B**: JFT-3B is an internal Google dataset and a larger version of the JFT-300M. It has over 3 billion images that have been annotated with a class structure of around 30k labels using a semi-automated procedure. Also, the dataset and the pre-trained weights are **not** open-source.

- **WIT-400M**: WIT-400M is a dataset that contains 400 million web image-text pairs, and is used to train CLIP [15]. CLIP does not release the dataset, but made all of the pre-trained models **available** [3]. In this paper, we utilize the CLIP-pretrained models in our experiments.

### B.2 Visual encoder architectures

In this paper, we use the visual encoder and textual encoder as shown in Table 3 and 4.

---

[1] https://www.image-net.org
[2] https://github.com/facebookresearch/vmz
[3] https://github.com/openai/CLIP

Table 3: CLIP-ResNet hyperparameters

| Model | Embedding dimension | Input resolution | ResNet blocks | width | Text Transformer layers | width | heads |
|---|---|---|---|---|---|---|---|
| RN50 | 1024 | 224 | (3, 4, 6, 3) | 2048 | 12 | 512 | 8 |

Table 4: CLIP-ViT hyperparameters

| Model | Embedding dimension | Input resolution | Vision Transformer layers | width | heads | Text Transformer layers | width | heads |
|---|---|---|---|---|---|---|---|---|
| ViT-B/32 | 512 | 224 | 12 | 768 | 12 | 12 | 512 | 8 |
| ViT-B/16 | 512 | 224 | 12 | 768 | 12 | 12 | 512 | 8 |
| ViT-L/14 | 768 | 224 | 24 | 1024 | 16 | 12 | 768 | 12 |
| ViT-L/14-336px | 768 | 336 | 24 | 1024 | 16 | 12 | 768 | 12 |

## B.3 Batch Gather for Distributed InfoNCE

Instead of Data-Parallel Training (DP), which is single-process, multi-thread, and only works on a single machine, Distributed Data-Parallel Training (DDP) is a widely adopted single-program multiple-data training paradigm for single- and multi-machine training. Due to GIL contention across threads, per-iteration replicated model, and additional overhead introduced by scattering inputs and gathering outputs, DP is usually slower than DDP even on a single machine.

---

**Algorithm 1:** Numpy-like Pseudocode that illustrates the role of Batch Gather in Distributed InfoNCE.

```
# text_encoder: encoder network for text input
# vision_encoder: encoder network for vision input, e.g., images or videos.
# V: minibatch of vision inputs
# T: minibatch of text inputs
# N: the local batch size of each GPU, e.g.,16
# M: the number of GPUs, e.g.,8
# N * M: the global batch size for multi-gpu training, e.g.,128

# extract feature representations of each modality
local_vision_features = vision_encoder(V) # shape: [N, embed_dim]
local_text_features = text_encoder(T) # shape: [N, embed_dim]

# normalization
local_vision_features = l2_normalize(local_vision_features, axis=1)
local_text_features = l2_normalize(local_text_features, axis=1)

# batch_gather is a function gathering and concatenating the tensors across GPUs.
all_vision_features = batch_gather(local_vision_features) # shape: [N * M, embed_dim]
all_text_features = batch_gather(local_text_features) # shape: [N * M, embed_dim]

# scaled pairwise cosine similarities
# shape = [N, N * M]
logits_per_image = logit_scale * image_features @ all_text_features.t()
# shape = [N, N * M]
logits_per_text = logit_scale * text_features @ all_image_features.t()

# The logits are then used as inputs for N*M-way (e.g., 128-way) classification,
# resulting in a loss value corresponding to N inputs in each GPU.
# Then Distributed Data Parallel mechanism takes care of averaging these across GPUs,
# which becomes equivalent to calculating the loss over NMxNM (e.g.,128x128) similarities.
```

---

Hence, we develop the Distributed InfoNCE based on DDP for large batch size and fast training. The core of the Distributed InfoNCE implementation is batch gathering. Say there are M GPUs and each GPU gets N input pairs, we need to calculate the $NM \times NM$ similarity matrix across the GPUs for InfoNCE loss. Without batch gathering, each GPU only computes a local $N \times N$ matrix, *s.t.* $N \ll NM$, Then the cosine similarity and the InfoNCE loss would be calculated only for the pairs within a single GPU and later their gradients would be averaged and synced. That's obviously not what we want.

The batch gathering for Distributed InfoNCE is presented as follows. When calculating the similarity matrix (and thus the logit scores across text inputs for each image/video), a GPU only needs to hold M vision features, and perform matrix product with NM text features, yielding an M×NM matrix. This computation is distributed (*i.e.*, sharded) across N GPUs, and we have calculated NM×NM similarities across the GPUs in total. The loss we employ is symmetric and the same happens *w.r.t.* text inputs. As shown in Algorithm 1, we also give an example pseudocode to help you understand the statement.

## B.4   Text template

In Table4 of the submission, we study several text input forms, including class names, single hard template, multiple hard templates, and learnable templates. More details are as follows:

**Class name**   To build textual embeddings, we utilize the category names of the dataset as the text input, *e.g.*, *"eating hotdog"*, *"driving car"*, *etc*.

**Single hard template**   We employ the hand-crafted template *"a video of a person {class name}."* to form a sentence as input.

**Multiple hard templates**   CLIP [4] provides 28 templates for Kinetics, one of which is the above single template. We use these multiple templates as the text augmentation during training. At each iteration, we choose one template at random as text input. Then, using the above single hard template as input, we perform the evaluation.

**Learnable templates**   We adopt the automated prompt CoOp [16] to describe a prompt's context using a set of learnable vectors. Specifically, the prompt given to the text encoder is designed with the following form,

$$t = [\text{V}]_1 [\text{V}]_2 \dots [\text{V}]_M [\text{class name}], \tag{1}$$

where each $[\text{V}]_m$ ($m \in \{1, \dots, M\}$) is a vector of the same size as word embeddings, and $M$ is a hyperparameter indicating the number of context tokens. We set the $M$ to 4.