# OpenReview forum: "Transferring Textual Knowledge for Visual Recognition"
_NeurIPS.cc/2022/Conference — NeurIPS 2022 Submitted_

### Official Review · Reviewer_yU63 · 2022-07-11

**Rating:** 5
**Confidence:** 4
**Soundness:** 3 good
**Presentation:** 1 poor
**Contribution:** 3 good

**Summary:**

The paper tackles the problem of transferring knowledge from pre-trained vision-language models to other downstream tasks. It proposes a new paradigm that makes use of the pre-trained text embedding to initialize the weights of the linear classifier head when testing on downstream tasks. Finally, the authors test their approach on Visual recognition and Image recognition.

**Questions:**

1. Any insights on why DistilBERT performs the same as CLIP in Table 1?

2. Why there is no comparison with other methods on the Image Recognition task?



**Limitations:**

There is no discussion related to the societal impact

**Strengths And Weaknesses:**

The paper tackles an important problem and seems to have promising results.

I find that the paper is quite confusing and hard to read. I would suggest to re-write some parts by specifying clearly what is the input, output of the model and what aims to be learnt or not (Section 2, lines 169-181, but not only limited to this).

Also, please be more specific about the downstream task and clearly specify what is the goal of the task, the input and output of the model.

Suggestions
Add some visual highlights to the tables (bold, italic, etc) so that the viewer knows what to look at.

---

> ### Author Response · Authors · 2022-08-01
> **The introduction of video recognition task and we will revise paper accordingly**
>
> Thank you for your comments.  We would like to answer your questions.
>
> ---
>
> > Q1: I find that the paper is quite confusing and hard to read. I would suggest to re-write some parts by specifying clearly what is the input, output of the model and what aims to be learnt or not (Section 2, lines 169-181, but not only limited to this).
>
> > Also, please be more specific about the downstream task and clearly specify what is the goal of the task, the input and output of the model.
>
> ---
> A1: Sorry for the confusion. **Video recognition** is our downstream task, which takes a video as input, and then fed into a learned model to estimate the action category of the video. The default pipeline of the video recognition is described as follows.
>
>
> * `Input:` The input has a size of 8x224x224x3 for 8 frames sampled from the video.
> * `Video encoder:` The input above is fed into the learnable visual encoder to get the video embedding (e.g., the size of 1x512).
> * `Output:` The model's output is a vector (size: 1x400) which provides the prediction value for each class. Specifically, the video embedding (size: 1x512) from the video encoder is passed to a classifier (size: 400x512) to produce the output vector.
>
>
> **The learnable part**: The classifier in our paradigm is intialized from the textual embedding of the class names and then frozen (fixed), leaving only the parameters in the **video encoder** to be learned. Our novelty is in appropriately initializing the classifier.
>
>
> ---
>
> > Q2: Suggestions Add some visual highlights to the tables (bold, italic, etc) so that the viewer knows what to look at.
>
> A2: Thank you for your suggestions. We will highlight the key results of the tables in the revision.
>
> ---
> > Q3: Any insights on why DistilBERT performs the same as CLIP in Table 1?
>
> A3: Both DistillBERT and CLIP are pre-trained with large-scale data, so they both have strong language modeling capabilities and can generate **good semantic targets**. Although the good semantic targets generated by DistillBERT are not aligned with the visual features of CLIP, it is easy to fit them with trainable visual encoders. Our observations in the experiment can also validate this, the loss of DistillBERT will be higher than CLIP in the early stage, but it will quickly decrease to the same level.
>
>
>
> ---
>
> > Q4: Why there is no comparison with other methods on the Image Recognition task?
>
> A4: In the ImageNet experiment of the original submission, our main purpose is to show that our approach can significantly speed up model convergence, so we can train the ViT-L/14 only takes 10 epochs to achieve 87.12%. Notably, instead of a large epoch (i.e., 300 epochs), complex augmentation (i.e., Gaussian Blur, Solarization, MixUp, etc) and various optimization tricks, we train models on ImageNet with only 10 epochs and the most basic augmentation (i.e., Random Crop).
>
> Here, we show the results of ViT-L model trained with our method for 10/30 epochs compared with other methods trained with 300 epochs as follows.
>
>
> | Method | Top1  | Epoch |
> | --- | --- | --- |
> | DeiT | 84.9% | 300 |
> | MLP-Mixer | 85.3% | 300 |
> | Meta Knowledge Distillation | 86.5% |  600 |
> | **Ours** | **87.1%** | **10** |
> | **Ours** | **87.9%** | **30** |
>
> ---
>
> As the major concerns come from what the video recognition task is, the missing visual highlights of experiment Tables, and comparisons on image recognition, we respectfully ask the reviewer to consider increasing the score if satisfied with the explanations above.

---

> > ### Comment · Reviewer_yU63 · 2022-08-07
> > **Thanks for the response**
> >
> > Hi,
> >
> > The provided comments mainly answer my concerns and provide more clarity. Please include all the details in the revised paper. Based on the response, I raise my score to Borderline Accept.

---

> > > ### Author Response · Authors · 2022-08-08
> > > **Appreciation**
> > >
> > > Dear Reviewer yU63:
> > >
> > > We are glad that our responses addressed your concerns and provided more clarity!
> > >
> > > Thanks for your recognition of our work. Please let us know if you have any unclear parts of our work on the last day.
> > >
> > > Best,
> > >
> > > Paper 2329 Authors.

---

> ### Author Response · Authors · 2022-08-07
> **Gentle Reminder**
>
> Dear reviewer yU63:
>
> We thank you for the precious review time and valuable comments. We have provided corresponding responses and results, which we believe have covered your concerns. We hope to further discuss with you whether or not your concerns have been addressed. Please let us know if you still have any unclear parts of our work.
>
> Best,
>
> Paper 2329 Authors.

---

### Official Review · Reviewer_L6WZ · 2022-07-12

**Rating:** 6
**Confidence:** 3
**Soundness:** 3 good
**Presentation:** 2 fair
**Contribution:** 2 fair

**Summary:**

This work proposes a method of transferring vision-language pre-training models to downstream visual recognition tasks. To do so, they propose to initialize the final linear classification layer with corresponding embedding values from a pretrained clip model, generating embedding by passing the class label as a prompt through CLIP. The classification layer is frozen after initialization, forcing the vision pipeline to optimize around the pretrained embeddings. Experiments demonstrate improvements in video and few-shot image recognition downstream tasks.

**Questions:**

- It would be great if the authors could clarify the LDA formulation, as it wasn't very clear to me. Is a vision-only model first trained on the data, after which LDA is fit, then used to initialize W for a newly trained vision model?

**Limitations:**

limitations not discussed

**Strengths And Weaknesses:**

strengths
- Proposed method is simple yet an effective means of repurposing trained embeddings for a different dowstream task
- Large improvement in experimental settings over standard pipeline.
- Notably, they also propose a strong baseline based on linear discriminant analysis

weaknesses
- There's some prior work transferring text embeddings from vision-language models to visual recognition tasks: [1]
- The description of the LDA formulation was a bit difficult to follow. It references a pretrained model on a certain training split, but it's not clear which model and which dataset this is referring to.
- The authors should probably verify whether there's any overlap in training data between the pretrained CLIP model and the dowstream task's data.
- For correctness' sake, I don't believe it's accurate to state the dimensionality reduction as part of the LDA algorithm, but rather something that can be done within the LDA framework.
- If dimensionality reduction is performed with LDA, it would be necessary to ablate whether the results would be better or worse without it, as the overall results seem really close to that of the proposed initialization method.




[1] Gupta et al. Aligned Image-Word Representations Improve Inductive Transfer Across Vision-Language Tasks. ICCV2017

---

> ### Author Response · Authors · 2022-08-02
> **Clarification on some misunderstandings**
>
> Thank you for your comments. We would like to answer your concerns.
>
> ---
>
> >  Q1: There's some prior work transferring text embeddings from vision-language models to visual recognition tasks: [1]
>
> > [1] Gupta et al. Aligned Image-Word Representations Improve Inductive Transfer Across Vision-Language Tasks. ICCV2017
>
>
> A1: Thanks for letting us know about this paper. We will cite it in the revision. Although the method described in [1] employs a learning objective similar to ours, we are addressing different problems: [1] is working on jointly learning aligned features embeddings for 3 specified tasks (i.e., object recognition, attribute recognition, and visual question answering), they embed the text labels using Word2vec. While we are focusing on:
>
> 1) how to properly finetune visual recognition tasks from the large-scale pre-trained image-text model.
>
> 2) exploring what types of the fixed classifier are optimal and examining how to leverage the inter-class correlation among the semantic information of different action categories.
>
>
>
> ---
>
> > Q2: The description of the LDA formulation was a bit difficult to follow. It references a pretrained model on a certain training split, but it's not clear which model and which dataset this is referring to.
>
> A2: The details are provided in Section4.2 (line 202-206). We use the **CLIP-pretrained visual encoder** to extract video embeddings of training split on the **Kinetics-400 dataset**.
>
> ---
>
> > Q3: Is a vision-only model first trained on the data, after which LDA is fit, then used to initialize W for a newly trained vision model?
>
>
> A3: We directly use the official CLIP-pretrained visual encoder to extract video embeddings, and the visual encoder is `not finetuned` on Kinetics-400. Then we perform LDA on the pre-extracted video embeddings of the training set in Kinetics-400 to initialize W and freeze it for finetuning the visual encoder on the Kinetics-400 dataset.
>
>
>
> ---
>
> > Q4: The authors should probably verify whether there's any overlap in training data between the pretrained CLIP model and the dowstream task's data.
>
>
> A4: In this paper, we mainly focus on the video recognition task with the Kinetics dataset. As shown in Fig.17 of CLIP official paper, CLIP has done the data overlap analysis on the Kinetics-700 dataset. They observe that there are less than 1% overlaps and many overlaps on Kinetics-700 are in fact all black transition frames. Then they conduct the experiment on overlapping data. The results show that the Kinetics-700 has no performance improvement, and even has an apparent 20% accuracy drop on the overlapping data.
>
>
> ---
>
> > Q5: For correctness' sake, I don't believe it's accurate to state the dimensionality reduction as part of the LDA algorithm, but rather something that can be done within the LDA framework.
>
>
> A5: Thanks for the comment! First, our previous version states that "LDA first projects the feature vectors into a lower dimension space" because of the first sentence "One way to view a linear classification model is in terms of dimensionality reduction." from the LDA introduction in chapter 4.1.4, page 186 of the renowned textbook [6].
>
> To avoid being controversial, we will change the sentence "Intuitively, the LDA first projects the feature vectors into a lower dimension space that maximizes the inter-class covariance and then estimates the likelihood of a sample to the class distributions." to "Intuitively, the LDA simultaneously maximizes the inter-class covariance and minimizes intra-class covariance."
>
>
> [2] CM Bishop. Pattern recognition and machine learning. Vol. 4. No. 4. New York: springer, 2006.
>
> ---
>
>
> > Q6: If dimensionality reduction is performed with LDA, it would be necessary to ablate whether the results would be better or worse without it, as the overall results seem really close to that of the proposed initialization method.
>
>
> A6: Sorry for the confusion. LDA is commonly used for feature classification or feature dimensionality reduction. However, in this work, we only use LDA for `feature classification` (in order to get "discriminant coefficients" as the classifier) instead of feature dimensionality reduction.
>
>
> For better understanding, we show the code which generates the LDA coefficient and `there is no dimension reduction`.
>
> ```
> import numpy as np
> from sklearn.discriminant_analysis import LinearDiscriminantAnalysis as LDA
> input = np.load('feats_labels_400class.npz')
> feats = input['feats']  # size: [24000, 512]
> labels = input['labels']  # size: [24000,]
> lda = LDA()
> lda.fit(feats, labels)
> classifier = lda.coef_ # size: [400, 512]
> ```
>
> ---
>
> As the major concerns come from the confusion about LDA classifier, we respectfully ask the reviewer to consider increasing the score if satisfied with the explanations above. Please do not hesitate to contact us if there are other clarifications or experiments we can offer.

---

> > ### Comment · Reviewer_L6WZ · 2022-08-08
> > **Thank you for your response**
> >
> > Thank you for your detailed responses. I believe all my questions have been clarified and will update my review accordingly.

---

> > > ### Author Response · Authors · 2022-08-08
> > > **Appreciation**
> > >
> > > Dear Reviewer L6WZ:
> > >
> > > We are glad that our responses addressed all your concerns and resolved the questions! !
> > > Thank you for updating your review accordingly. Please let us know if you have any unclear parts of our work on the last day.
> > >
> > >
> > >
> > > A minor note — There may be something wrong with the system since the score has not been updated yet. Could you kindly edit the review again?
> > >
> > > Best,
> > >
> > > Paper 2329 Authors.

---

> > > ### Author Response · Authors · 2022-08-09
> > > **Gentle Reminder**
> > >
> > > Dear Reviewer L6WZ:
> > >
> > > We are glad that our responses addressed all your concerns and resolved the questions!  And you said you will update your review accordingly.
> > >
> > > However, we observe that the score has not been updated yet.
> > >
> > > Just a friendly reminder that the deadline for updating your review is in two hours.
> > >
> > >
> > > Best,
> > >
> > > Paper 2329 Authors.

---

> ### Author Response · Authors · 2022-08-07
> **Gentle Reminder**
>
> Dear reviewer L6WZ:
>
> We thank you for the precious review time and valuable comments. We have provided corresponding responses and results, which we believe have covered your concerns. We hope to further discuss with you whether or not your concerns have been addressed. Please let us know if you still have any unclear parts of our work. Since there is only one day left in the discussion window, we sincerely hope that you would not miss our response.
>
> Best,
>
> Paper 2329 Authors.

---

### Official Review · Reviewer_KzRg · 2022-07-12

**Rating:** 7
**Confidence:** 5
**Soundness:** 4 excellent
**Presentation:** 4 excellent
**Contribution:** 4 excellent

**Summary:**

This paper revises the role of widely used classifiers from a novel perspective that has been overlooked. Based on the vision-language pre-trained model, this work introduces text priors into the standard recognition framework in a fixed-classifier fashion without additional training cost. A variety of different fixed classifiers are also discussed, and qualitative and quantitative results are provided. This work presents a simple yet effective paradigm for visual recognition. Prior to this, the traditional recognition paradigm often used pre-trained visual encoders with a randomly initialized classifier to finetune downstream recognition models without considering pre-trained text encoders. This study showed how properly using visual-language pre-trained models can significantly improve recognition performance and speed up model convergence.


**Questions:**

Please refer to my questions in the weakness part for details.

**Limitations:**

Yes.

**Strengths And Weaknesses:**

Strengths:
This work democratizes training on large-scale video/image datasets (i.e., Kinetics400, ImageNet) to a certain extent, achieving good accuracy with only a few epochs and also performing well in few-shot/zero-shot scenarios. As far as I know, it seems to be the first work to achieve 87+% on Kinetics400 using only publicly available pre-trained models, indicating that it is reproducible and can inspire future works. And it even performs better than works that used large-scale pre-trained models (e.g., JFT-3B/JFT-300M/FLD-900M) that were never open source.

Weakness:
- The authors may include more discussion about the computational cost and efficiency issues.
- What are the training details for few-shot video recognition?
- The authors may include the study of failure cases to help the readers to understand the limitation of this work.
- What's the benefit of using randomized orthogonal matrix?

---

> ### Author Response · Authors · 2022-08-02
> **More detailed explanation**
>
> Thank you for your appreciation of this work.
>
> ---
>
> > Q1: The authors may include more discussion about the computational cost and efficiency issues.
>
> A1: Thank you for your suggestion.  In the following table, we present the computational cost and efficiency of our models, where **"vid/s"** represents the average number of videos per second. The larger "vid/s" represents higher efficiency. We follow the common inference settings by using a single NVIDIA A100 GPU to measure the throughput. We use a batch size of 16 to measure the throughput.
>
> Our models achieve the superior throughput and less FLOPs comparing with previous transformer-based methods on Kinetics-400 dataset.
>
>
>
>
>
> | Method | Frames | Top1 | FLOPs | Params | Throughput|
> | --- | --- | --- | --- | --- | --- |
> | ViViT-L/16-320 | 8  | 81.3% | 3992G | 310.8M | 4.2 vid/s |
> | **Ours ViT-B/32** | 8  | 78.5% | 23.7G | 71.6M | 322.5 vid/s |
> | **Ours ViT-B/16** | 8  | 81.5% | 90.3G | 69.9M | 126.5 vid/s |
> | **Ours ViT-L/14** | 8  | 85.4% | 415.4G | 230.4M | 35.5 vid/s |
>
>
>
> ---
>
> > Q2: What are the training details for few-shot video recognition?
>
> A2: Details are provided in Section 4.3 line 268-274. In this paper, we study a more challenging K-shot All-way situation. For the results in Table 8, we scale the few-shot task up to categorize all the action categories in the three dataset (i.e., Kinetics400, UCF51, HMDB51) with just one sample (e.g., K=1) per category for training. All training details are the same as all-shot video recognition (those in other experiments) except for more epochs (i.e.,200 for few-shot video recognition while 30 for other experiments).
>
>
> ---
>
> > Q3: The authors may include the study of failure cases to help the readers to understand the limitation of this work.
>
> A3: We will discuss the following limitation in the revision:
>
> When the annotated category labels do not contain semantic information, their textual features will obviously not contain semantic knowledge. For example, if the class labels are numerical values such as 0, 1, 2, etc., rather than semantic descriptions such as "eating burger", "eating cake", "eating chips", etc., in this case, the textual classifier will not contain knowledge, and it is better to use the LDA classifier.
>
>
> ---
>
> > Q4: What's the benefit of using randomized orthogonal matrix?
>
> A4:
> 1) We would like to clarify that randomized orthogonal matrix is just one of the four possible initialization methods. Randomized orthogonal matrix  is not advocated. Our proposed initialization is the forth, which is Textual embedding vectors.
> 2) Benefits of randomized orthogonal matrix: We remove the inter-class correlation of classifier by using randomized orthogonal matrix. As expected, this initialization has inferior performance.

---

> > ### Comment · Reviewer_KzRg · 2022-08-07
> > **Thanks for the authors' reply.**
> >
> > Thank you for thoroughly addressing my concerns. My concerns are covered by the responses.
> > I would like to encourage the authors to include the results of efficiency (in A1) in the final revision, which demonstrates the superior efficiency of the model. Thanks for the limitation in A3, I think the response makes sense and is reasonable.
> > After reading through this and other responses, the extra experimental results and explanations make the paper better. In my opinion, this paper is simple yet effective, provides significant performance, and could serve as a new standard for more future works in the video recognition field. I would like to raise my confidence to 5 and keep my rating at 7-accept.

---

> > > ### Author Response · Authors · 2022-08-07
> > > **Thanks for your recognition of our work**
> > >
> > > Dear reviewer KzRg:
> > >
> > > Thank you for your reply!
> > > We will include the limitation and result of efficiency and in our final version.
> > >
> > > We really appreciate your recognition of this work!
> > >
> > >
> > >
> > > Best,
> > >
> > > Paper 2329 Authors.

---

### Meta-Review · Area_Chair_AXpZ · 2022-08-27

**Recommendation:** Reject
**Confidence:** Certain

**Metareview:**

The paper aims to study the idea of transferring textual knowledge from vision-language pertained models to visual recognition or specifically the adaption of CLIP for downstream visual recognition tasks. The authors proposed to revise the role of the linear classifier and replace the classifier with the embedded language representations of the object categories. The idea is simple (and somewhat trivial) and authors demonstrated some promising results in experiments. Despite the positive aspects, there are several major concerns with this paper: 1) the technical depth of the method is weak (the paper only made a minor change to the paradigm of using vision-language pretrained model), 2) the novelty of the idea is limited, in fact the idea of transferring text knowledge or zero-shot/few-shot adaption of CLIP for downstream visual recognition tasks has been extensively studied in CLIP and many its variants, but there lack of comparisons with those work.  3) the empirical study is not convincing and comparison are not extensive (many CLIP variants and related baselines are not compared); also the related work was poorly written with many missing related work in recent advances of CLIP and video related CLIP variants. Overall, the paper has some interesting simple idea that may be worth for further investigation but the paper is not strong enough for publication.

**Award:**

No

---

### Decision · Program_Chairs · 2022-09-14

Reject